

# High-resolution seismic reflection surveying to delineate shallow subsurface geological structures in the karst area of Shenzhen, China

Zhihui Wang[1,2,3], Christopher Juhlin[2,], Qingtian Lü[1], Xiaoming Ruan[1], Zhendong Liu[1], Chenghua Yu[4], Mingchun Chen[5]

[1]Chinese Academy of Geological Sciences, Beijing, 100037, China
[2]Department of Earth Sciences, Uppsala University, Uppsala, 75236, Sweden
[3]Key Laboratory of Geological Safety of Coastal Urban Underground Space, Ministry of Natural Resources, Qingdao, 266121, China
[4]Shenzhen Investigation & Research Institute CO., LTD, Shenzhen, 518026, China
[5]Sinopec Geophysical Corporation Nanfang Branch, Chengdu, 610200, China

*Correspondence to*: Qingtian Lü (lqt@cags.ac.cn)

**Abstract.** Near-surface seismic reflection surveys can produce high-resolution geological structural images for engineering and hydrological investigations. To help delineate shallow subsurface complex geological structures in a karst area and to better understand limestone cave formation, a high-resolution 2D seismic reflection profile was acquired and processed in the urban area of Shenzhen, China. The stacked images detail subsurface structures down to depths of 80-90 m, including a concave shaped reflection, two thrusts and one normal fault, as well as a hard rock basement reflection at the southern end of the profile which could not be mapped by borehole investigations due to the limited drilling depth. Our interpretations correlate well with borehole data and synthetic modeling. Limestone caves are mainly found along faults and near rivers in this area. Our results provide new insight on the formation mechanism and distribution of the karst caves. Movement along faults and erosion generated fractures and fissures in the limestone that provide channels for rainwater and groundwater to circulate. These waters, rich in carbonic acid, dissolve minerals in the limestone, resulting in the formation of underground cavities. Mapping of the subsurface with geophysical methods can contribute to mitigation of karst hazards in the Pingshan district, Shenzhen.

**Keywords**

Seismic reflection; Karst; Sinkhole; Subsurface structure; Geological hazards

## 1 Introduction

Karst landscapes, characterized by their unique geological formations shaped by dissolution of soluble rocks, such as limestone and dolomite, are renowned for their beauty and ecological significance. However, beneath the surface hidden dangers are present. Karst hazards pose challenges for various industries, particularly those dependent on stable ground conditions, such as construction, agriculture, and infrastructure development. Losses due to the fast-acting nature of karst encounters can be direct (e.g., human casualties and damage to property), indirect (e.g., interruption to businesses, transport



infrastructure and communication networks) or intangible, especially if they occur in areas of high population density (Galve et al., 2012; Bobrowsky, 2013; Intrieri et al., 2015; Sevil et al., 2017; Pazzi et al., 2018).

In southeast China, buried karst, with high fissure water content, high permeability and variable shapes, is widely distributed (Cui et al., 2015). Shenzhen, a world-scale metropolis, located in the southern part of the Guangdong province, has significant areas of Carboniferous rock distributed in the Longgang, Pingshan and Dapeng districts where karst features are a hazard. During metro, railway and building construction, karst and karst caves at depths from 2 m to 50 m are found in dolomite, dolomitic limestone, dolomitic marble, marble, crystalline limestone and breccia limestone (Li et al., 2011; He et al., 2020). Its presence challenges the construction of tunnels and use of shield tunneling machines. For example, the disturbance produced by a shield tunneling machine may induce the ceiling of a karst cave to collapse, with water present in the cave damaging the machine and constructed tunnel. Furthermore, diaphragm wall collapse, water or mud ingression, ground collapse and long-term instability are often encountered in karst regions (Cui et al., 2015). Currently, many direct and indirect techniques to detect buried karst and karst caves have been proposed (Lolcama et al., 2002; Hoover, 2003; Kaufmann and Romanov, 2009; Song et al., 2012; Kaufmann, 2014; Samyn et al., 2014; Putiška et al., 2014; Kaufmann and Romanov, 2016; Sevil et al., 2017; Pazzi et al., 2018; Hussain et al., 2020; Wang et al., 2020; Muzirafuti et al., 2020; Yordkayhun, 2021; Stan-Kłeczek et al., 2022; Yordkayhun et al., 2022; Liu et al., 2023). Among the direct methods, drilling and electric cone penetration tests are the most common and useful. On the other hand, indirect techniques can be employed to delineate subsurface karst size and distribution and extrapolate borehole data to a wider area. Geophysical techniques based on a physical contrast between a cave and the surrounding rocks provide an economical and non-invasive alternative, or complement, for mapping geological structures, and are often used in attempts to detect the presence of karst caves and voids below the surface. Methods include seismic reflection/refraction, multichannel analysis of surface waves (MASW), the H/V spectral ratio method, electrical resistivity tomography (ERT), induced polarization (IP), self-potential (SP), ground penetrating radar (GPR) and micro-gravity. In recent years, with a focus on improving resolution, efficiency, and cost-effectiveness (Juhlin et al., 2010; Hardage et al., 2011; Martinez et al., 2011; Lundberg et al., 2014; Shan et al., 2014; Brodic et al., 2015; Malehmir et al., 2016; Pugin and Yilmaz, 2017; Bharadwaj et al., 2017; Yilmaz, 2021; Sun et al., 2022; Pertuz and Malehmir, 2023), the high-resolution seismic reflection method for shallow exploration and imaging of local subsurface heterogeneities has emerged as a powerful technique to identify and map near surface geological structures with good precision and depth penetration.

In this study, we delineate the near surface geological structures in a karst area with high-density and -resolution seismic reflection data. Previous investigations at the site relied mainly on boreholes for the mapping and no analysis was performed on how karst caves may form in the area. Our seismic results are correlated with borehole data, synthetic modeling and ground-penetrating radar data. The seismic reflection images provide new insight in understanding the formation mechanism and distribution of karst and karst caves.



## 2 Geological setting and physical properties

Geologically, Shezhen is located in the South China Block southwest of the Zhenghe-Dapu fault. Multi-stage, complex formation of folds and faults and intensive metamorphism took place in the pre-Caledonian, Caledonian, and Hercynian to Indosinian orogenies (Cui et al., 2015). The conspicuous Lianhuashan fault zone in Shenzhen is interpreted as the southern extension of the lithosphere-scale Zhenghe-Dapu fault zone, considered as the boundary between the late Mesozoic Coastal terrane and the early Paleozoic Wuyi-Yunkai orogen (Li et al., 2020). The sedimentary units affected by the Lianhuashan

fault zone include Paleozoic and Early to Middle Jurassic rocks. Due to significant magmatic activity, granite and igneous rocks that formed during the Yanshanian period are the dominant rock type. The Yanshanian orogeny involved tectonic movement that mainly included block orogeny and the development deep faults and wide folds, along with significant intrusions of granitic magma, massive acidic volcanic eruptions, and magmatism-related mineralization.

On a local scale (Fig. 1), the Paleozoic sequence comprises Silurian-Carboniferous sandstone, shale, slate, limestone,

siltstone, gristone and coal seams (Shenzhen Geology, 2009), that are distributed along the NE-striking and NW-dipping Paotaishan-Hengtougang transpressional fault (PHF) and the Shijingling-Huangzhukeng transpressional fault (SHF). Holocene alluviums and flood plain deposits affected by the PHF and SHF consist of gravel, silty sand, sandy clay, silty clay and other unconsolidated deposits. Medium-grained granites are present northwest of the study area with U-Pb zircon crystallization ages of about 108 Ma (Shenzhen Geology, 2009). The seismic survey line is located on the Holocene flood

plain deposits unit, as shown by the red rectangular area in Figure 1.





**Figure 1: Regional geologic map of the Pingshan district, Shenzhen. Red rectangle (Fig. 3a) shows study area location (after Shenzhen Geology, 2009).**



The geological section consists of 19 engineering geology boreholes that provide more detailed information on lithology down to 70 m (Fig. 2). There are six main geological units from the top to the bottom of the section, including fill soil, sand, silty clay, weathered sandstone and shale, and limestone. Limestone caves were found in boreholes BH52 and in BH56 to BH59 in the vicinity of the interface between weathered shale and limestone. The fill soil has a relatively constant thickness of c. 10 m in all boreholes. The thickness of the sand layer increases from north to south, however, weathered shale shows

the opposite trend. Weathered sandstone is only present from BH46 to BH51. Limestone is present from BH46 to BH59 and absent from BH60 to BH64. Thickness variations of the different sedimentary deposits in the horizontal direction suggest that multi-stage tectonic events occurred in this area. The top of the weathered shale has a concave upward shape between BH53 to BH57. Based on borehole sampling, the groundwater table is present at the top of the sand layer. Due to the different compositions of the inner shale, differential weathering is suggested to have occurred at the southern end as

indicated in BH60, and shown in Figure 2b, a potential source for seismic reflections.

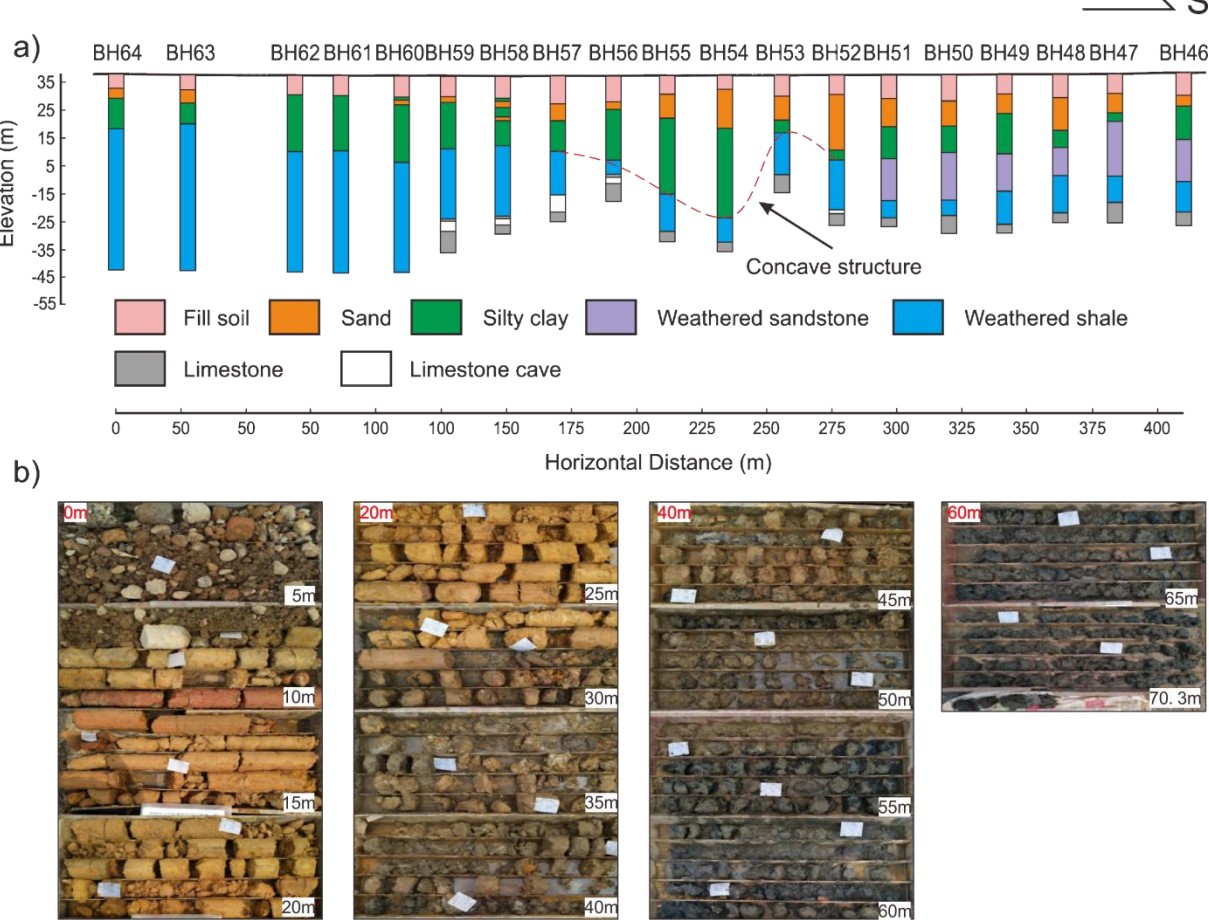

Figure 2: Borehole lithology and representative core photos. (a) Borehole section consists of 19 boreholes down to a maximum depth of 70 m, located about 20 m away from the seismic survey line. (b) core box images from BH62 borehole, 0-6.9m, fill soil; 6.9-27.3m, silty clay; 27.3-70.3m, weathered shale, some weathered into clay and some are still hard rocks.





Compressional and shear velocity and density data collected in the neighborhood of study area provide information on the
physical properties of the subsurface soils and rocks (Table 1), and can help identify where reflection impedance changes
may be located. Fill soil with 1.55 g/cm³ has the lowest density and limestone with 2.65 g/cm³ has the highest value, the
density of sand with gravel is a somewhat higher than silty clay and other weathered rocks. Compressional and shear
velocities have similar variational trends, however, compressional velocity is 3-4 times faster than shear velocity. Limestone
compressional velocity is up to 3125 m/s; sand with gravel and silty clay have the lowest compressional velocity. A four-
layer seismic reflection model is suggested in this area based on the velocity and density data. The first layer consists of fill
soil; the second is made up of gravel and silty clay; the third is completely and strongly weathered rocks, including
sandstone and shale; the fourth is limestone.

**Table 1: Compressional velocity (Vp), shear velocity (Vs) and density of soil and rock collected from the adjacent area, Vp and Vs
from logging velocity and density from geotechnical testing of borehole samples.**

| Lithology | Density (kg/m³) | Vs (m/s) | | | Vp(m/s) | | |
|---|---|---|---|---|---|---|---|
| | | min | avg. | max | min | avg. | max |
| Fill soil | 1550 | 283 | 283 | 283 | 1157 | 1157 | 1157 |
| sand with gravel | 2040 | 212 | 212 | 212 | 891 | 891 | 891 |
| silty clay | 1940 | 209 | 215 | 219 | 878 | 944 | 1002 |
| sandstone, completely weathered | 1870 | 412 | 428 | 443 | 1423 | 1434 | 1445 |
| sandstone, strongly weathered | 1890 | 541 | 587 | 687 | 1654 | 1810 | 1960 |
| shale, strongly weathered | 1960 | 482 | 482 | 482 | 1532 | 1532 | 1532 |
| limestone, moderately weathered | 2650 | 1346 | 1360 | 1374 | 3060 | 3092 | 3125 |

## 3 Data acquisition and geometry

High-resolution seismic data were acquired in November 2022 in the Pingshan district of Shenzhen, southeast China (Fig. 1).
The survey line is located along Longping Road and lies between light rail transit line 1 and the Ciao river (Fig. 3a, c). The
surface topography from south to north varies smoothly with about an elevation difference of 2 m. The geological section
from the 19 boreholes allows a comparison with the seismic results, however, they were drilled at an elevation c. 5 m higher
than the seismic survey line (Fig. 3c) and 20 m offset from it. A 5 kg sledgehammer with a 4 cm thick metal plate was used
as the seismic source along with 5 Hz SmartSolo 3C nodal units for recording. The seismic data were acquired using five
segments. Every segment consisted of 148 units with a fixed geometry of 1 m receiver spacing and 2 m source spacing with
74 units overlapping (Fig. 3b). The total length of the survey line is 417 m. A sampling rate of 1 ms was used and 1000 ms
of data were retrieved from the nodes for each source point. Table 2 shows the seismic acquisition parameters. Three raw
shot gathers recorded at different locations along the survey line show some of the characteristics of the seismic wavefield
(Fig. 4). Direct waves, surface waves, reflections and air waves are present in all gathers. Reflections dominate in the time
window of 50-150 ms, shallower reflections are masked by surface waves, air waves and direct waves at the near offsets.







**Figure 3: (a) Locations of receivers (cyan triangles), shots (red stars) and boreholes (green dots), (b) Geometry of the seismic acquisition profile, receiver spacing 1 m, shot spacing 2 m, separated into five segments, each segment consisted of 148 fixed units with 74 units overlapping, (c) Field work photo of the seismic survey line (cyan line) and geological section consisting of 19 boreholes (red line), the geological section is 5 m higher than the seismic survey line (the aerial image from © Google Maps).**





**Table 2: Acquisition parameters for the high-density and -resolution seismic data in the municipality of Shenzhen, southeast of China.**

| Attribute | Parameters |
|---|---|
| Recording system | Smart Solo |
| Receiver | IGU-16HR 3C, 5Hz |
| Source | 5 kg sledgehammer |
| Receiver interval | 1 m |
| Shot interval | 2 m |
| CMP interval | 1 m |
| Sampling rate | 1 ms |
| Recording length | 1 s |
| Minimum offset | 0 m |
| Maximum offset | 147 m |
| Survey geometry | Asymmetric split spread, roll-along |
| Number of receivers | 417 |
| Number of shots | 209 |

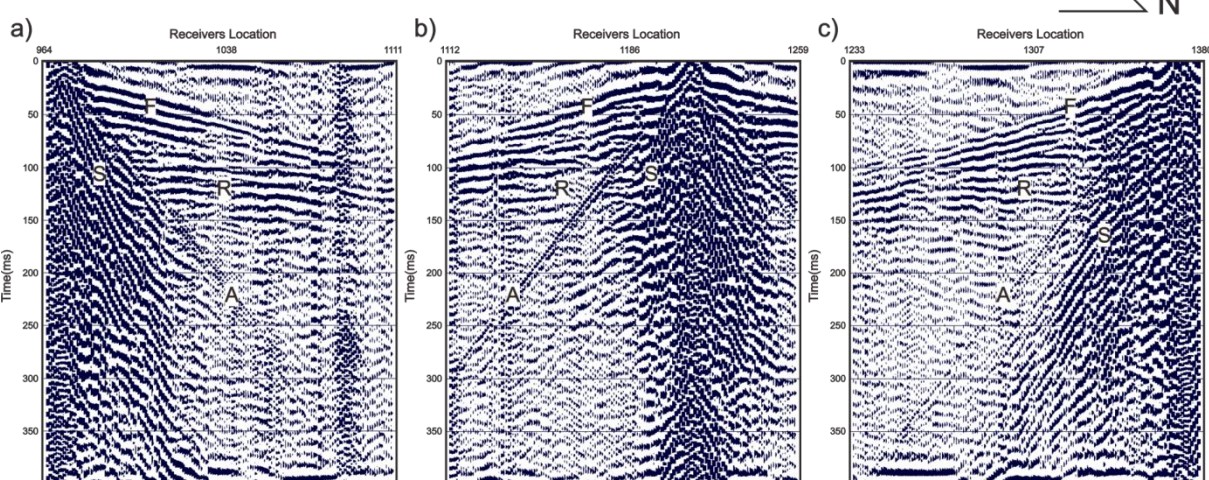

**Figure 4: Three typical shot gathers from different locations along the seismic survey line: (a) located in the south (shot No. 968), center (shot No. 1214) and north (shot No. 1374). Reflections from the time window of 50-150 ms are notable, shallower reflections are masked by surface waves, air waves and direct waves at the near offsets, F, S, R and A represent first breaks, surface waves, reflections and air waves, respectively.**

## 4 Data processing

Data processing followed a standard workflow to improve the signal-to-noise ratio and resolution after stacking vertically repeated shot records (twice at each shot location). Tomographic refraction statics were applied to account for traveltime variations in the very near surface, while band-pass filtering, spectral equalization and linear noise suppression were applied



to remove different types of noise. An iterative velocity analysis was performed to obtain the best velocity model and continuity of reflections. The continuity was improved further through surface-consistent residual static corrections. After stacking, band-pass filtering and f-x domain deconvolution were used to further reduce random noise and enhance the coherency of reflections. Post-migration were applied to move dipping reflectors to their true subsurface positions and

collapse diffractions. Finally, time to depth conversion with the smoothed NMO velocity field was performed to obtain an approximate depth model and to help in interpreting geological structures (Fig. 9). Processing steps and parameters are shown in Table 3, while two important steps are discussed in detail below.

**Table 3: Processing workflow and parameters for the high-density and -resolution seismic data in the municipality of Shenzhen, southeast of China.**

| Step | Processing workflow |
|:---:|:---:|
| 1 | Data input, read SEGD format data from the tape and convert it to SEGY format |
| 2 | Merged data |
| 3 | Trace edits, kill noisy traces |
| 4 | Geometry, add shots and receivers coordinate and calculate CMP binning 1 m |
| 5 | First breaks, automatic and manual picking |
| 6 | Tomography statics |
| 7 | Band-pass filter, 50-60-180-220 Hz |
| 8 | Spectral equalization, 50-80-200-220 Hz |
| 9 | Median filter, airwaves 340 m/s, surface waves 250 m/s, linear noise 600 m/s |
| 10 | Surgical muting, surgical top mute above first breaks |
| 11 | 1st Velocity analysis, 0 ms-500 m/s, 2000ms-4000 m/s |
| 12 | Residual statics |
| 13 | 2nd Velocity analysis, 0 ms-1000 m/s, 200ms-2000 m/s |
| 14 | Automatic gain control, 150ms |
| 15 | NMO, 70% stretch |
| 16 | Stack |
| 17 | Band-pass filter, 60-80-180-200 Hz |
| 18 | F-X deconvolution |
| 19 | Migration |
| 20 | Time to depth conversion, velocity field from smoothed NMO velocity model |



## 4.1 Noise attenuation

Raw shot gather recordings are dominated by low frequency and strong amplitude noise, such as surface waves and coherent noise. A typical power spectrum shows that the data contain frequencies in the range 5-440 Hz (Fig. 5a). To suppress the low frequencies and strong amplitude noise, a band-pass filter with corner frequencies of 50-80-180-220 Hz was applied to the raw data (Fig. 5b). Direct waves, refractions, reflections, air waves and surface waves are recognized with apparent velocities of approximately 1000 m/s, 1800 m/s, 2200 m/s, 340 m/s and 220 m/s, separately. After band-pass filtering, a spectrum equalization filter was used to further reduce noise and enhance the weaker amplitude signals. Compared with Fig. 5b, two sets of reflections are notable in Fig. 5c and marked with arrows. Finally, the air waves, surface waves and other linear noise were attenuated by median filters at velocities of 340 m/s, 250 m/s and 600 m/s, respectively. Reflections are now clearly observed in Fig. 5d.

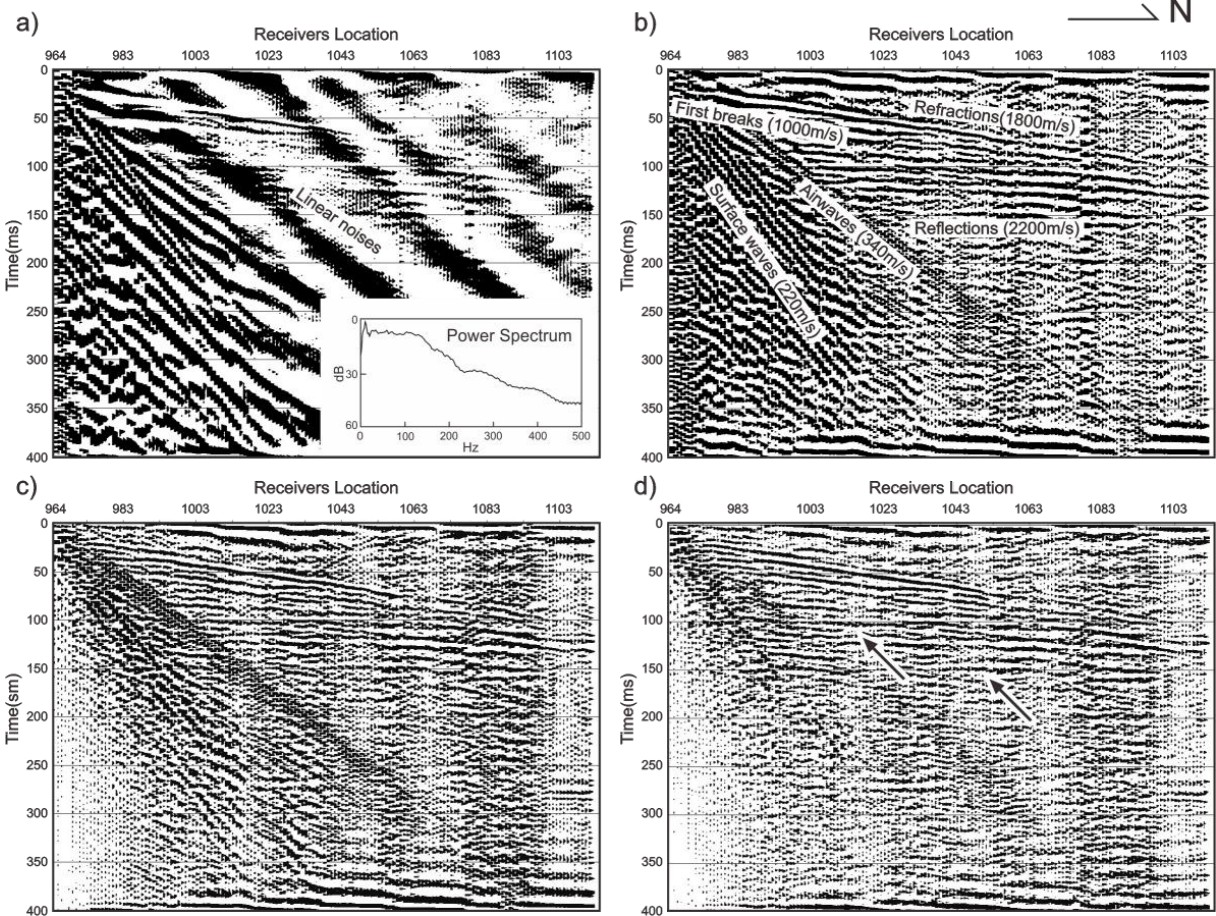

**Figure 5: Noise attenuation processing steps for the No. 964 shot gather. (a) Raw shot gather and its power spectrum; (b) after a band-pass filter (50-80-180-220 Hz), typical apparent velocities for direct waves, refractions, reflections, air waves and surface waves are about 1000 m/s, 1800 m/s, 2200 m/s, 340 m/s and 220 m/s, respectively; (c) after spectrum equalization filter (50-60-200-220Hz); (d) after median filtering to remove noise, air wave 340 m/s, surface wave 250 m/s, linear noise 600 m/s.**




## 4.2 First break traveltime tomography

First break picking and traveltime tomography were performed to correct for the seismic wave traveltime delays in the very near surface low velocity zone and build a P-wave velocity model. Based on the picked first breaks, a two-layer model was chosen as the initial velocity model, with apparent velocities of about 700 m/s and 1650 m/s (Fig. 6a). The ray density plot shows an even distribution in most regions except at about 100-240 m along the profile and at elevations of 20-28 m (Fig. 6b). Root-mean-square errors are approximately 2.4 ms after the fifth iteration (Fig. 6c), indicating a stable result has been attained. The inverted velocity model suggests a two-layer structure over the depth of investigation, the top one with low P-wave velocity is consistent with the fill soil layer, the bottom one with a high P-wave velocity may be correlated to the sand-silty clay layer (Fig. 6d).

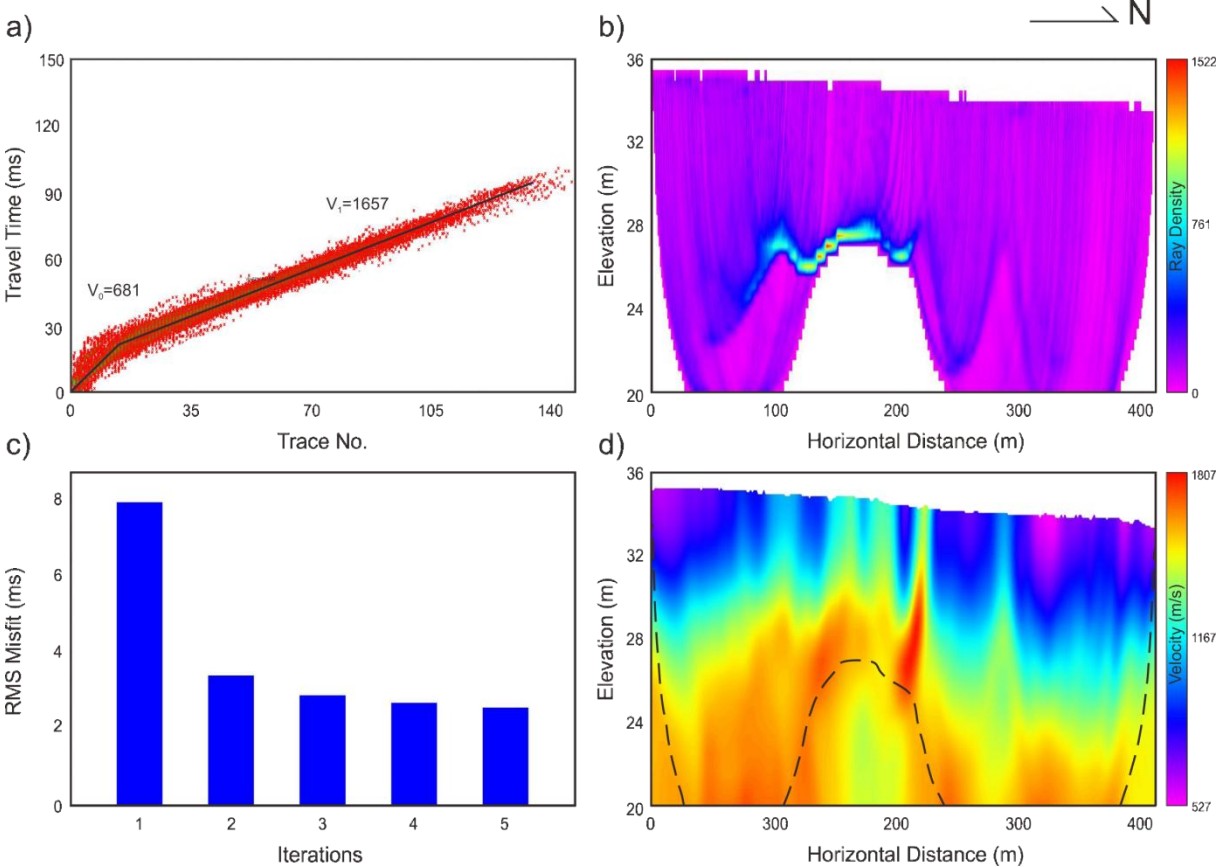

**Figure 6: Traveltime tomography for first breaks. (a) First break traveltimes for all shots, a two-layer model with apparent velocities of 681 m/s and 1667 m/s was taken as the initial velocity model; (b) Ray density model of 20-36 m above sea level; (c) Root mean square (RMS) errors for different iterations; (d) Inverted P-wave model, the black dashed lines represent no rays in the area and velocity is interpolated below it.**



## 5 Synthetic modeling

Synthetic modeling was carried out using the 2D elastic wave equation. A four-layer model (Fig. 7a) with fill soil (Layer 1), sand and silty clay (Layer 2), weathered shale (Layer 3) and limestone (Layer 4) was used to simulate the subsurface geological structures. The model parameters for the compressional velocity, shear velocity and density are based on the values in Table 1 and first break velocities of the compressional and shear waves using the vertical and radial components. The model length is the same as for the real data and the depth is to set to 300 m to mitigate reflections from the bottom of

the model. A Ricker wavelet with a central frequency of 70 Hz was used as a source. In order to be consistent with the field data, the geometry and recording parameters were identical with Figure 3b and Table 2.

Figure 7b shows one synthetic shot gather, which has the same location as the real gather in Figure 5a-d. The R3 reflection is generated from the top of the limestone and matches well the real data gather. Reverse time migration was applied using the CREWES Matlab Toolbox after stack (Fig. 7c). The reflections from the top of sand and silty clay (R1), weathered shale (R2)

and limestone (R3) are clearly visible. The result also suggests the presence of a concave shaped structure consistent with the borehole section (Fig. 2a) and real data (Fig. 8). At a distance 0-100 m along the profile and a time of c. 100 ms, a strong amplitude reflection indicates the limestone which is not mapped by the borehole data due to the limited borehole depth. This reflection is clear in the real data and provides added constraints on the geological structure of the area (Fig. 8).





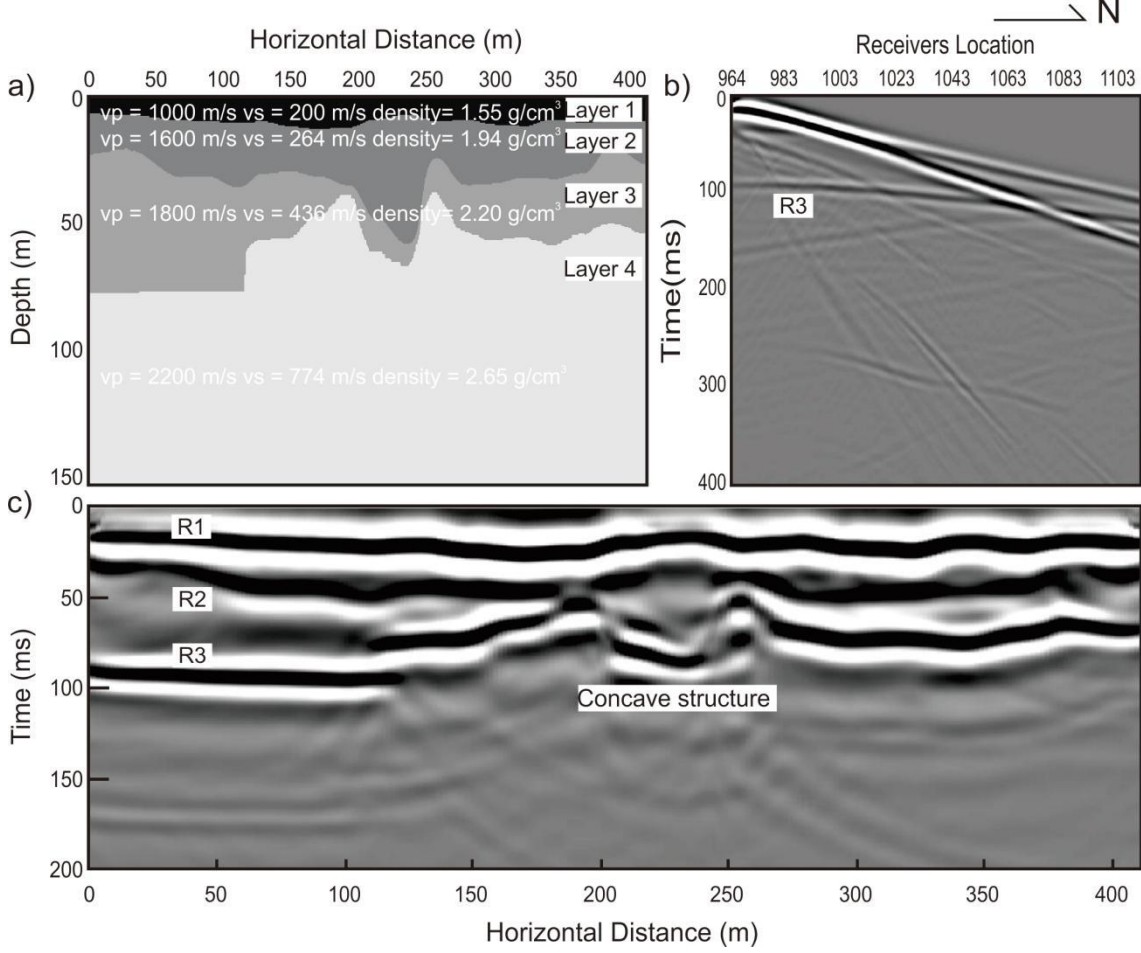

Figure 7. Synthetic model and results. (a) the model of P-wave, S-wave and density based on borehole data and field raw data, Layer 1, Layer 2, Layer 3 and Layer 4 represent fill soil, sand and silty clay, weathered shale and limestone, respectively, and can be correlated with borehole data. (b) synthetic shot gather at shot No. 964, the same location as Fig. 5a - d. (c) Post-stack time migration section.

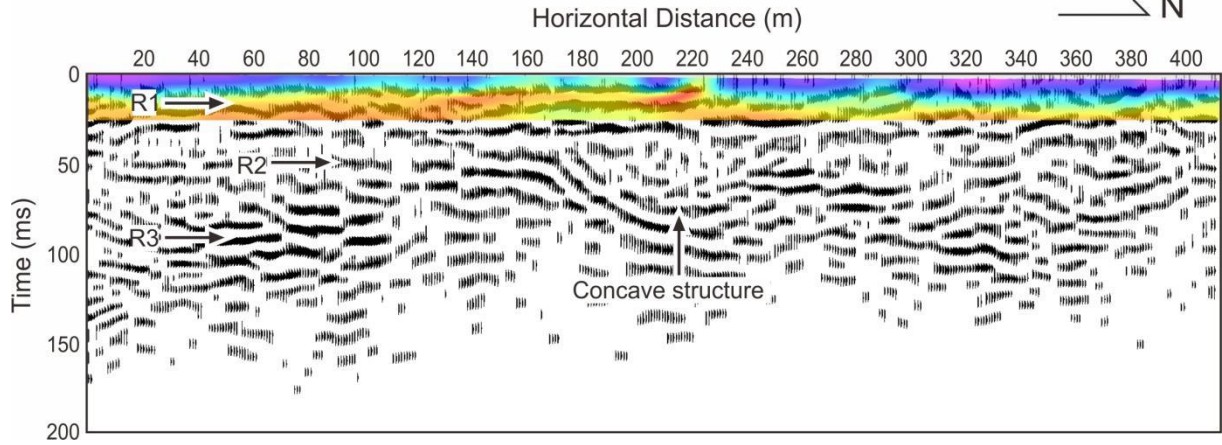



**Figure 8: Seismic reflection migrated image overlaid with the time converted tomography result, an apparent discontinuity of reflections and velocity at horizontal distance 220 m is observed. R1 and R2 represent reflections from the top and bottom of the sand and silty clay layer, respectively; R3 originates from the top of the limestone. At horizontal distance 220 m and time 75 ms, a concave shaped structure exists that correlates with the borehole section.**

## 6 Results and discussion

### 6.1 Reflection characteristics from the migrateed seismic section

Reflection seismic processing steps focused on removing random and coherent noise and imaging high-frequency reflections in order to resolve the near-surface geological structure. The seismic tomography image suggests that the uppermost reflection horizon on the real data can be interpreted as the fill soil – sand and silty clay interface (Fig. 8). Furthermore, the physical property sample analyses indicated that three seismic interfaces can be mapped in the area and we interpret our seismic section accordingly. In addition to the top of the sand and silty clay unit (R1) two other sets of reflections are imaged down to a depths of 80-90 m. These, labeled R2 and R3, represent reflections from the weathered shale and the bedrock interface, consistent with the borehole data and synthetic modeling. Between R2 and R3, some reflections are produced by clay from the weathered and unweathered shale, as shown in Fig. 2b. The R1 reflection is distorted at a distance of 220 m. When compared to the near surface velocity model from first break tomography (Fig. 8), there is also a variation in velocity at this location. This suggests that the thickness of fill soil varies due to paleotopography or fault slipping. However, it is not possible at present to determine which factor is correct. We have interpreted three faults in the middle part of the seismic section, F1, F2 and F3 (Fig. 9). Seismic reflections show apparent offset at the locations of these suggested faults and together they form the upward concave structure seen in the seismic image, as well as in the seismic modeling. The locations also correlate with where karst caves are found in the borehole section. If correctly interpreted as faults, this indicates a connection between the karst caves and faulting in this area.

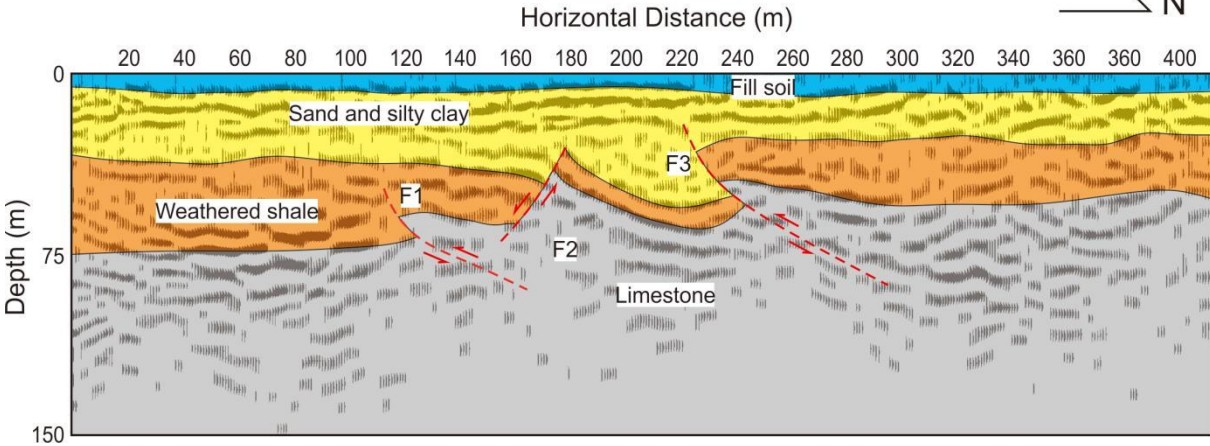

**Figure 9. Geological interpretation of the depth converted seismic reflection section. Four-layer model as indicated by the seismic data with fill soil, sand and silty clay, weathered shale and then limestone. Two thrusts and one normal fault are shown and labeled by F1, F3 and F2, respectively.**





**Figure 10. Karst distribution in the Panshan district, Shenzhen. Karst is not only correlated with carbonate rocks, but also faults and rivers (after Shenzhen Geology, 2009).**



## 6.2 Formation mechanism and distribution area of karst caves

The seismic reflection section combined with the borehole data provide information on the detailed subsurface geological
structure in the area. They may also help in understanding the formation and distribution of karst caves in this area. Previous
studies suggest that at least four stages of regional scale tectonic and magmatic events have taken place in South China (Shu
et al., 2014), forming sets of transpressional fault structures, N to NW dipping (Fig. 1). As shown in Fig. 9, two thrusts
labeled as F1 and F3 and one normal fault (F2) are suggested to be present in the study area. Slipping along these faults and
erosion generated fractures and fissures in the limestone probably provided channels for rainwater and groundwater to enter
the subsurface from the Ciao river (Fig. 3c). Carbonic acid in the water dissolved the limestone until it became saturated with
carbonate minerals. As saturated groundwater flowed away from the area, unsaturated underground water flowed in and
dissolved further the limestone. Finally, underground cavities were formed. This may give a reasonable explanation why
limestone caves developed along the faults and near the rivers in the Pingshan district, Shenzhen (Fig. 10).

## 7 Conclusions

A 417 m long high-resolution seismic reflection profile was acquired to image shallow subsurface geological structure in a
karst area. Notably, four layers with fill soil, sand and silty clay, weathered shale and the bedrock are imaged in high
resolution on the stacked seismic section. Also, three small-scale faults are delineated which are closely related to karst caves.
To validate the seismic image, synthetic modeling was conducted to compare with the real data and help in the geological
interpretation. Furthermore, a near surface velocity model from first break tomography, coincident with a borehole section
20 m away from the seismic line, correlates well with the seismic reflection image and confirms the reliability of it.
Integrating the geology and geophysics results in the delineation of the fine subsurface geological structure and provides a
better understanding of the formation process and spatial distribution of karst caves in the Pingshan district, Shenzhen.
Multi-stage regional scale tectonic and magmatic events produced cracks, fractures and fissures in the limestone that formed
channels where rainwater and groundwater with carbonic acid could circulate and form caves and voids. This interpretation
provides valuable insight into the development of karst caves and the research contributes to the mitigation of karst hazards.

## Data availability

To request the data associated with this research, contact the corresponding author of the article after the publication of this
work.



**Author contribution**

ZW and QL conceptualized and designed this study. ZW, ZL and XR were involved in the data acquisition. ZW, CJ, ZL, XR and MC were responsible for the data processing. CY provided boreholes data. ZW, CJ and QL led the geological interpretation. ZW wrote the initial draft and CJ reviewed it. All authors participated to the results discussion and approved the submission of this paper.

**Competing interests**

The authors acknowledge that there are no conflicts of interest.

**Acknowledgments**

This research was funded by the National Natural Science Foundation of China (42174119) and China Geological Survey Project (DD20211314). We acknowledge researchers and students from Chinese Academy of Geological Sciences who contributed to the data acquisition and Professor Hemin Koyi and Alireza Malehmir from Uppsala University for their

helpful discussion.

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
