# Peer review of "High-resolution seismic reflection surveying to delineate shallow subsurface geological structures in the karst area of Shenzhen, China"

_EGUsphere, 2024_

## Referee Comment (RC1)

Review for manuscript egusphere-2024-4050 „High-resolution seismic reflection surveying to delineate shallow subsurface geological structures in the karst area of Shenzhen, China"

**General comments**

In their study, the authors aim to image the near subsurface in the urban area of Shenzhen with reflection seismic and boreholes in order to better understand the local karst features.

Overall, the manuscript is very interesting and fits to the scope of the journal. The manuscript has a good structure and is mostly well written in terms of language. Nonetheless, there are some points that should be revised/adjusted. Therefore, my recommendation is a publication of the manuscript after a minor revision.

In the following, I will give a chapter-by-chapter explanation for my decision, and I will try to give the authors helpful tips on how to improve the manuscript.

**Abstract**

General comment: I suggest that the authors should write a little bit more about the actual seismic results.

Page 1, lines 15-16: "The stacked images detail subsurface structures down to depths of 80-90 m, including a concave shaped reflection …" → What are these concave shaped reflections? Give an interpretation.

Page 1, lines 17-18: "Our interpretations correlate well with borehole data and synthetic modeling." → What kind of synthetic modelling?

**Introduction**

General comment: A well-written introduction.

Page 2, lines 61-62: "Our seismic results are correlated with borehole data, synthetic modeling and ground-penetrating radar data." → What type of synthetic modelling?

**Geological setting and physical properties**

Page 3, line 65: "Shezhen" → Shenzhen

Page 3 line 72: "…and the development deep faults…" → development of deep faults

Page 3 line 75: "gristone" → gritstone

Page 6 line 100: "Compressional and shear velocity and density data collected in the neighborhood of study area…" → How was the data acquired? With which method? It is only mentioned in the table caption. Please also mention it in the text.

Page 6 line 102: "2.65 g/cm$^3$" → Use same units in text and table. In the table you use kg/m$^3$.

Figure 1:

The colours in the small overview map in the upper left corner seem to represent other geological units than in the large figure, although the colours are similar. Please change the colours in order to prevent confusion and add the explanation to the legend.

The abbrevations Dsh, DDh, $Qh^{al}$, $Qh^{fp}$, $C_{1C1}$ and γK1 are not explained.

In the legend, the blank spaces between words are varying in width even in the same row.

Table 1: Maybe you can also add a column with the $V_P/V_S$ ratio and discuss it later. This is also an interesting geotechnical parameter.

**Data acquisition and geometry**

Figure 3: In figure 3a, the names of the boreholes are barely visible. Please adjust it by maybe changing the colour of the font.

Figure 4: The labelling of the different wave types (F, S, R, A) is poorly visible in the figure. Maybe use a different font colour like red.

**Data processing**

Page 9, line 141: "…to remove different types of noise." → Specify that a little bit more.

Page 9, line 144: "Post-migration were applied…" → What type of post-migration?

**Noise attenuation**

Page 10, line 153: "…a band-pass filter with corner frequencies of 50-80-180-220 Hz was applied…" → Why did you choose this specific frequency range? Describe it.

Page 10, line 157: "…two sets of reflections are notable in Fig. 5c and marked with arrows." → The arrows are in figure 5d!

Figure 5:

In figure 5c, show also a power spectrum after spectral equalization.

In figure 5d, I can still see some remnants of the surface waves. This might impair the quality of the stacked section.

**Synthetic modelling**

Figure 7: The information about $V_P$, $V_S$, and density for the fourth unit in figure 7a is barely visible. Change the colour of the font.

**Results and discussion**

Page 14, line 205: "migrated" → migrated

General comment: The description of the result is okay, but a proper discussion is missing! You have to discuss your paper in the context of the field of research. You have to describe the meaning and relevance of your results. Furthermore, you have to compare your results with other regional studies, but, in the best case, also other studies carried out across the world. There are many other papers dealing with reflection seismic in the context of karst and also karst related to faults. Right now, you only cite and compare your results to only one paper. Besides that you should also discuss possible shortcomings of your research design, like the acquisition parameters and the processing scheme → critical analysis of the used methods.

> I to separate the results and discussion chapter and to write an extra chapter only dealing the proper discussion of the results and the methods.

---

## Author Response (AR1)

Dear Editors and Reviewers:

Thanks for your comments and professional suggestions concerning the manuscript entitled "High-resolution seismic reflection surveying to delineate shallow subsurface geological structures in the karst area of Shenzhen, China". Those comments and suggestions are all valuable for us to revise and improve our paper, as well as important for improving the readability of the paper. We have made corrections and improvements according to the comments and would provide our responses point by point as follows.

**Responses to community comments**

*Comment 1: Lines 28-29. "Karst hazards pose challenges for various industries, particularly those dependent on stable ground conditions, such as construction, agriculture, and infrastructure development". Insert recent papers that face the relationship between industry and karst environments.*

**Response 1:** We will add the two recent papers in the manuscript *(**see revised page 1 line 30**)*.

*Comment 2: Line 63. Please, clearly disclose the 3 to 4 specific objectives of your research by using numbers (e.g., i, ii, and iii).*

**Response 2:** The research specific objectives were disclosed in the revised manuscript, including i) Testing of the reflection seismic method over karst terrain; ii) Comparison of reflection seismic results with geotechnical drilling; iii) Integration of physical properties of samples with reflection seismic modeling; and iv) Delineating shallow subsurface complex geological structures in a karst area and understanding limestone cave formation better (**see revised page 2 lines 59-62**).

*Comment 3: "Zhenghe-Dapu fault zone". Is it a normal fault? Please, specify.*

**Response 3:** Multi-stage, complex formation of folds and faults and intensive metamorphism took place in the pre-Caledonian, Caledonian, and Hercynian to Indosinian orogenies in the southeast of China. The early activity of Zhenghe-Dapu fault zone is ductile shear, and the middle and late stages are thrust and mid-development detachment (**see revised page 3 lines 66-70**).

*Comment 4: Lines 65-75. Please, provide more detail on the carbonate stratigraphy.*

**Response 4:** More details on the carbonate stratigraphy including rock mass structure and lithology are provided in the revised manuscript (**see revised page 4 lines 83-88**) .

*Comment 5: Line 220. I can see in the interpretation of the seismic line thrusts and normal faults. You need to insert in the setting more information on either compressional or extensional tectonics to back-up the output of your seismic survey.*

**Response 5:** The thrusts and normal faults happened during compressional tectonics as far as we know. More information about tectonics in the region was provided in the revised manuscript (**see revised page 3 lines 68-70**).

*Comment 6: Lines 229-238. Please, provide more detail on the nature and the geometry of the karst landforms.*

**Response 6:** Thanks for your suggestions. We have provided more information on the karst landforms using references and the geometry of karst caves from boreholes (**see revised pages 14-15 lines 239-243**) .

**Comment 7:** *Line 266. Consider inserting recent literature on karst environments that has been suggested.*

**Response 7:** Thanks again. We added two recent papers on karst environments (**see page 18 line 299 and page 19 line 342**).

**Comment 8:** *Figure 1. Insert symbols for the type of fault.*

**Response 8:** We reproduced Figure 1 and made it more readable following your suggestion (**see page 3 line 78**).

**Comment 9:** *Figure 2a. All the tops of the wells have the same elevation. Possible? Please, check or specify that we are on a plain.*

**Response 9:** Yes, they are on a plain. The largest elevation difference is about 0.5 m.

**Comment 10:** *Figure 10. "Karst area". Specify the types of karst landforms.*

**Response 10:** We reproduced Figure 11 and specified the types of karst landforms in the revised version (**see page 16**).

**Comment 11:** *Figure 10. Specify the type of faults.*

**Response 11:** We reproduced Figure 11 and specified the types of faults in the revised version (**see page 16**).

**Responses to referee comments #1**

**Comment 1: Abstract**

General comment: I suggest that the authors should write a little bit more about the actual seismic results.

Page 1, lines 15-16: "The stacked images detail subsurface structures down to depths of 80-90 m, including a concave shaped reflection …"→ What are these concave shaped reflections? Give an interpretation.

Page 1, lines 17-18: "Our interpretations correlate well with borehole data and synthetic modeling."→ What kind of synthetic modelling?

**Responses 1:** Thanks a lot. We have given more actual seismic results in the Abstract, including: (1) Concave shaped reflections mean seismic reflection events curved inwards in Figure 8; (2) We carried out synthetic reflection seismic modelling using the 2D elastic wave equation to simulate the seismic wavefield to help us give a comparison with the actual data and used the results to help interpret the geology (**see page 1 lines 16, 18 and 19**).

**Comment 2: Introduction**

General comment: A well-written introduction.

Page 2, lines 61-62: "Our seismic results are correlated with borehole data, synthetic modeling and ground-penetrating radar data." → What type of synthetic modelling?

**Responses 2:** We carried out reflection seismic modeling using the 2D elastic wave equation to simulate the seismic wavefield to help us give comparison to actual data and geological interpretation (**see page 2 line 63**).

*Comment 3:* **Geological setting and physical properties**

Page 3, line 65: "Shezhen" → Shenzhen

Page 3 line 72: "…and the development deep faults…"→ development of deep faults

Page 3 line 75: "gristone"→ gritstone

Page 6 line 100: "Compressional and shear velocity and density data collected in the neighborhood of study area…"→ How was the data acquired? With which method? It is only mentioned in the table caption. Please also mention it in the text.

Page 6 line 102: "2.65 g/cm$^3$" → Use same units in text and table. In the table you use kg/m$^3$

Figure 1:

The colours in the small overview map in the upper left corner seem to represent other geological units than in the large figure, although the colours are similar. Please change the colours in order to prevent confusion and add the explanation to the legend.

The abbrevations Dsh, DDh, Qhal, Qhfp, C1C1 and γK1 are not explained.

In the legend, the blank spaces between words are varying in width even in the same row.

Table 1: Maybe you can also add a column with the VP/VS ratio and discuss it later. This is also an interesting geotechnical parameter.

**Responses 3:** Thanks for pointing out the spelling mistakes. We have corrected them (**see page 3 lines 66, 75 and 82**).

Compressional and shear velocity were acquired by velocity logging and the density was measured by the volumetric cylinder method in the lab (**see page 5 lines 107-109**).

We corrected density unit of "2.65 g/cm$^3$"to 2650 kg/m$^3$, same with Table 1 (**see page 5 lines 109-110**).

We have revised Figure 1 and Table 1 and made them clear and readable in the revised manuscript (**see page 3 and 5**).

*Comment 4: Data acquisition and geometry*

Figure 3: In figure 3a, the names of the boreholes are barely visible. Please adjust it by maybe changing the colour of the font.

Figure 4: The labelling of the different wave types (F, S, R, A) is poorly visible in the figure. Maybe use a

different font colour like red.

**Responses 4:** Figure 3 and Figure 4 are improved in the revised manuscript (**see pages 7 and 8**).

*Comment 5: Data processing*

Page 9, line 141: "…to remove different types of noise."      Specify that a little bit more.

Page 9, line 144: "Post-migration were applied…"      What type of post-migration?

**Responses 5:** We gave more information about filtering and migration (**see page 9 line 151**).

*Comment 6: Noise attenuation*

Page 10, line 153:"…a band-pass filter with corner frequencies of 50-80-180-220 Hz was applied…"→
Why did you choose this specific frequency range? Describe it.

Page 10, line 157:"…two sets of reflections are notable in Fig. 5c and marked with arrows."   The
arrows are in figure 5d!

Figure 5:

In figure 5c, show also a power spectrum after spectral equalization.

In figure 5d, I can still see some remnants of the surface waves. This might impair the

quality of the stacked section.

**Responses 6:** We carried out different frequency range tests and then determined the best frequency
range to be 50-80-180-220 Hz (**see page 10 line 166**).

Arrows and a power spectrum have been added in Figure 5c in the revised manuscript (**see page 10**).

Actually, there are still some remnants of the surface waves. Because it is difficult to eliminate surface
wave completely. Stacking further filters out surface waves.

*Comment 7:* **Synthetic modelling**

Figure 7: The information about VP, VS, and density for the fourth unit in figure 7a is barely visible.
Change the colour of the font.

**Responses 7:** We have changed the font colour in Figure 7a in the revised manuscript (**see page 13
line 207**).

*Comment 8: Results and discussion*

Page 14, line 205: "migrateed"→migrated

General comment: The description of the result is okay, but a proper discussion is missing!

You have to discuss your paper in the context of the field of research. You have to describe the
meaning and relevance of your results. Furthermore, you have to compare your results with other
regional studies, but, in the best case, also other studies carried out across the world. There are many
other papers dealing with reflection seismic in the context of karst and also karst related to faults.
Right now, you only cite and compare your results to only one paper. Besides that you should also

discuss possible shortcomings of your research design, like the acquisition parameters and the processing scheme→critical analysis of the used methods.

I to separate the results and discussion chapter and to write an extra chapter only dealing the proper discussion of the results and the methods.

**Responses 8:** Thanks again. We have corrected the spelling mistake in the revised manuscript (**see page 14 line 218**). Meanwhile, the discussion has been modified and improved (**see pages 14-16 lines 239-265**).

**Responses to referee comments #2**

*Comment 1: In the introduction Authors claim: "The seismic reflection images provide new insight in understanding the formation mechanism and distribution of karst and karst caves", however a discussion on the broader implications of the study that would enhance its scientific contribution is lacking.*

**Response 1:** We have presented a schematic model of karst cave formation after analyzing and comparing with other results across the world which contribute to understanding the distribution of karst caves (**see pages 14-16 lines 239-265**).

*Comment 2: What specific new insights do you believe your study provides beyond previous research?*

**Response 2:** Previous research focused on the location of karst caves. We found that faults contributed to the formation mechanism and distribution of the karst caves. Movement along faults and erosion generated fractures and fissures in the limestone that provide channels for rainwater and groundwater to circulate. These waters, rich in carbonic acid, dissolve minerals in the limestone, resulting in the formation of underground cavities (**see page 15 lines 243-258**).

*Comment 3: Can your findings be generalized to other karst regions, or are they specific to the Shenzhen geological setting and this case? Especially that the covers only a single 417-meter profile, which may not fully capture regional variations in karst structures. Have you considered extending the survey to a larger area to validate the observed structures?*

**Response 3:** At present, we do not think our findings can be generalized to other karst regions, just specific to our research area. We agree with you that a 417-meter profile can not fully capture regional variations in karst structures. So we are looking for new funding to support our research and to extend the survey to a larger area to validate the observed structures in Shenzhen in in the future.

*Comment 4: Are you planning on analyzing the horizontal components from measured data? What possible additional insights could that bring into your study? Consider including it in the study.*

**Response 4:** Yes, we tried to process the radial and transverse components data, however, we did not obtain clear reflections from the horizontal components. Active and passive surface wave and H/V spectral ratio analyses were also carried out.

*Comment 5: How do you plan to integrate other geophysical methods in future studies to enhance the reliability of karst detection?*

**Response 5:** We will give a comparison and evaluate different seismic exploration methods on karst detection after integrating the surface wave data and using the H/V spectral ratio method. Integrated passive and active seismic data analysis may give more reliable results in our research area.

**Comment 6:** *Figure 1: try to compress the figure. Add the location mark of the Shenzhen region in a broader context.*

**Response 6:** Figure 1 has been improved in the revised manuscript (**see page 3**).

**Comment 7:** *Figure 2: are the borehole section elevations cropped to the same level? Or the elevation is so ideally constant?*

**Response 7:** All boreholes are on a plain. The largest elevation difference is about 0.5 m. So they are nearly at the same level in Figure 2.

**Comment 8:** *Try to limit the number of references in the introduction choosing the most relevant*

**Response 8:** We have removed some of the less relevant references in the revised manuscript (**see pages 18-20**).

Thank you very much for your attention and time. Do not hesitate to contact us if you have any questions about the revised manuscript. Looking forward to hearing from you.

Best Regards,

Zhihui Wang